# A closer look onto breast density with weakly supervised dense-tissue masks

**Mickael Tardy**[1,2]                            MICKAEL.TARDY@EC-NANTES.FR
[1] *LS2N UMR6004, Ecole Centrale Nantes, France,* [2] *Hera-MI, SAS, Nantes, France*

**Bruno Scheffer**[2,3]                         BRUNO.SCHEFFER@HERA-MI.COM
[3] *Institut de cancérologie de l'Ouest, Saint-Herblain, France*

**Diana Mateus**[1]                           DIANA.MATEUS@EC-NANTES.FR

## Abstract

This work focuses on the automatic quantification of the breast density from digital mammography imaging. Using only categorical image-wise labels we train a model capable of predicting continuous density percentage as well as providing a pixel wise support frit for the dense region. In particular we propose a weakly supervised loss linking the density percentage to the mask size.

**Keywords:** Deep Learning, Weakly supervised segmentation, Regression, Breast density

## 1. Purpose

Breast density is a biomarker for breast cancer development risk that suggests that the risk of cancer development increases with denser breasts. Moreover, the detection of cancer in dense tissues and, more generally, in dense breasts is often considered more challenging due to the similar visual aspects of normal and abnormal tissues, which complicates the interpretation of mammographic images. For the above reasons, we argue that computer-aided decision systems for early breast cancer detection should both, quantitatively evaluate the breast density, and evaluate the spatial distribution of the dense tissues.

## 2. Methods

In clinical practice, breast density is usually assessed image-wise using a classification grid like the BI-RADS (Breast imaging-reporting and data system) (Irshad et al., 2016). In the present work, we propose to estimate breast density at the pixel level while using only image-wise ground truth from the BI-RADS scale. Our goal is to generate a breast density mask, identifying pixels associated with the tissue that contributed to the density class. To achieve our goal, we propose a novel loss linking the sought breast density mask to the globally estimated breast density (fig. 1). We formulate the problem as a weakly supervised binary semantic segmentation. Our approach is related to recent efforts to reduce the amount of supervision (Carneiro et al., 2017; Dubost et al., 2017).

In practice, we rely on a modified U-Net architecture (Ronneberger et al., 2015) and on an extended 12-class density grid that improves the density resolution compared to traditional BI-RADS classification (4th edition). Compared to the state-of-the-art, our

classification and segmentation scheme does not rely on the model's attention but uses a loss function efficiently correlating a tissue mask with the target breast density values. Moreover, the output is constrained with the breast binary mask removing useless activations.

## 3. Results

The database for training and tests consists of 1232 and 370 images respectively. We got promising results with a mean absolute error (MAE) of 6.7% for the density regression estimate (see tab. 2) and an accuracy of 78% for 4-class BI-RADS density classification (tab. 1). Our comparison baseline is a VGG-like regression model trained on the same dataset.

To validate the segmentation performance, we collected regions of interest on several images (16) and calculated the $Dice = 0.65$. Overall we obtain clinically meaningful segmentation masks offering valuable insights into the spatial distribution of the dense tissues (fig. 2). In comparison, we demonstrate the inefficiency of the attention-based techniques for the breast density mask generation.

In addition we validate our approach on the INBreast (Moreira et al., 2012) database. Without any additional training and a simple preprocessing we obtained 65% accuracy and $MAE = 13\%$. We note that our results are comparable to other works on the same dataset (64.53%, (Schebesch et al.), 67.8% (Angelo et al., 2015).

## 4. Conclusions

Our approach to link breast density classification to the spatial distribution of dense tissue has a positive effect on classification scores while providing an additional output mask of the dense regions. These results are interesting given the considerably low requirements on ground truth (just a class instead of an image mask) and the size of the training dataset.

Table 1: 4-class BI-RADS classification performance. All models are trained with 12-class grid. $\mathcal{L}$ is the proposed loss.

| Model | Metrics | | | | |
|---|---|---|---|---|---|
| | Accuracy | Precision | Recall | $F_1$-score | Cohen kappa |
| VGG+Sigmoid+MSE | 0.764 | 0.782 | 0.764 | 0.766 | 0.891 |
| U-NET+Softmax+$\mathcal{L}$ | 0.684 | 0.729 | 0.684 | 0.679 | 0.838 |
| **U-NET+ReLU+$\mathcal{L}$** | **0.779** | **0.809** | **0.779** | **0.781** | **0.891** |

Table 2: Regression performances of the studied models. All models, except the last two are trained with 12-class grid. $\mathcal{L}$ is the proposed loss.

| Dataset | Metrics | | |
|---|---|---|---|
| | MAE (%) | MxAE (%) | C-index |
| VGG+Sigmoid+MSE | 6.545 | 31.964 | 0.820 |
| U-NET+Softmax+$\mathcal{L}$ | 8.303 | 34.404 | 0.789 |
| **U-NET+ReLU+$\mathcal{L}$** | **6.661** | **32.156** | **0.839** |

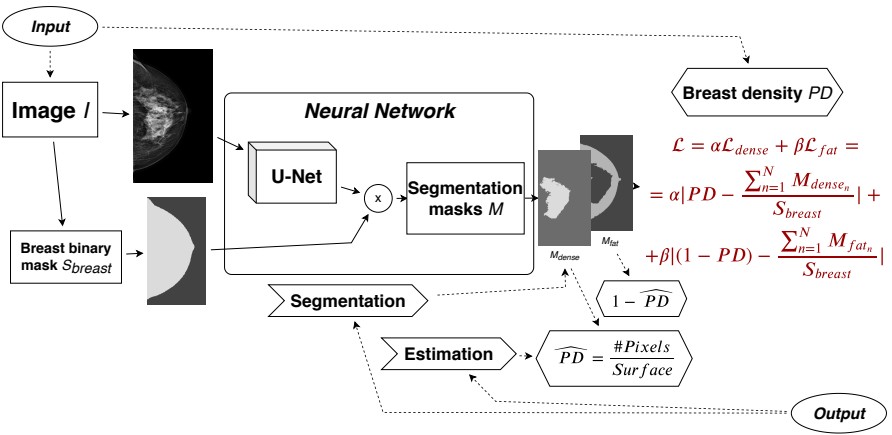

Figure 1: Proposed spatial distribution evaluation model. First input **Image I** is fed to a **U-Net network**, then, u-net output is combined with a **binary breast mask** $S_{breast}$ to yield the output **segmentation masks** $M = \{M_{dense}, M_{fat}\}$. The **combined loss** $\mathcal{L}$ guides the model training using image-wise $PD$ density label.

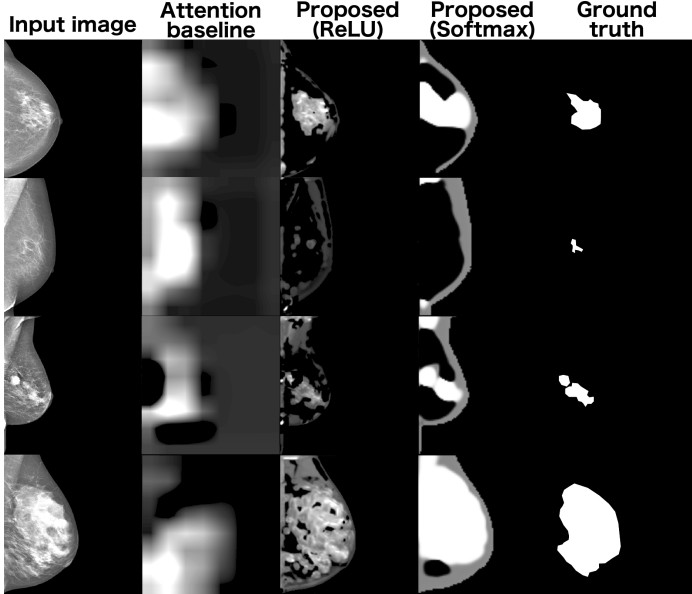

Figure 2: Resulting dense tissue masks. **First column**: input images, **second column**: activation masks produced by the attention-based baseline, **third column**: density masks $M_{dense}$ of ReLU-trained model, **fourth column**: density masks $M_{dense}$ of Softmax-trained model and **fifth column**: ground truth

## Acknowledgments

Research funding is provided by Hera-MI, SAS and Association Nationale de la Recherche et de la Technologie via CIFRE grant no. 2018/0308.

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
