# OpenReview forum: "A closer look onto breast density with weakly supervised dense-tissue masks"
_MIDL.io/2019/Conference/Abstract — MIDL Abstract 2019_

### Official Review · AnonReviewer2 · 2019-04-25
**interesting application of weakly supervised learning**

**Rating:** 3
**Confidence:** 3

**Review:**

This abstract presents a technique for weakly supervised learning of breast density with CNNs. This is a relevant application, because images with image-wide annotations are very abundant in mammography. The CNN produces a segmentation that is connected with (ground truth) a scalar density score through a predefined function. The results are competitive and visually compelling. I missed some more detail on the loss function, but this is understandable, given the space constraints.

---

### Official Review · AnonReviewer1 · 2019-04-30
**Weakly supervised approach for**

**Rating:** 3
**Confidence:** 2

**Review:**

The abstract presents a weakly supervised approach to do classification and obtain a segmentation mask.
The method is evaluated on two datasets for classification and regression task with the comparable results to works on the dataset.
The paper requires multiple readings to understand. It will help if the typos in the paper are corrected(Eg. In Table1. Baseline). In the extended version, clear motivation and description of loss function would be beneficial.

---

### Decision · Program_Chairs · 2019-05-06
**Acceptance Decision**

Accept